# Immigrant birds learn from socially observed differences in payoffs when their environment changes

**Michael Chimento** [1,2,3]*, **Gustavo Alarcón-Nieto** [3,4,5,6], **Lucy M. Aplin** [1,2,3,7]

**1** Department of Evolutionary Biology and Environmental Studies, University of Zurich, Zurich, Switzerland, **2** Centre for the Advanced Study of Collective Behaviour, University of Konstanz, Konstanz, Germany, **3** Cognitive and Cultural Ecology Research Group, Max Planck Institute of Animal Behavior, Radolfzell, Germany, **4** Department of Biology, Konstanz University, Konstanz, Germany, **5** Department of Migration, Max Planck Institute of Animal Behavior, Radolfzell, Germany, **6** International Max Planck Research School for Quantitative Behavior, Ecology and Evolution, Radolfzell, Germany, **7** Division of Ecology and Evolution, Research School of Biology, Australian National University, Canberra, Australia

* mchimento@ab.mpg.de

## Abstract

Longstanding theory predicts that strategic flexibility in when and how to use social information can help individuals make adaptive decisions, especially when environments are temporally or spatially variable. A short-term increase in reliance on social information under these conditions has been experimentally shown in primates, including humans, but whether this occurs in other taxa is unknown. We asked whether migration between spatially variable environments affected social information use with a large-scale cultural diffusion experiment with wild great tits (*Parus major*) in captivity, a small passerine bird that can socially learn novel behaviors. We simulated an immigration event where knowledgeable birds were exchanged between groups with opposing preferences for a socially learned foraging puzzle, living in similar or different environments. We found evidence that both immigrants and residents were influenced by social information and attended to the rewards that others received. Our analysis supported the use of a payoff-biased social learning by immigrants when both resources and habitat features were spatially variable. In contrast, immigrants relied more-so on individual learning when payoffs or the environment were unchanged. In summary, our results suggest that great tits assess the payoffs others receive and are more influenced by socially observed differences in payoffs when environmental cues differ in their new environment. Our results provide experimental support for the hypothesis that spatial variability is a strong driver for the evolution of social learning strategies.

**Data Availability Statement:** All data and code files are available from the Edmond database (https://doi.org/10.17617/3.FXC12W).

## Introduction

When animals move from one place to another through dispersal, migration, or nomadism, they often experience spatial variability, whereby environmental factors change across a

**Funding:** M.C. and L.M.A. were supported by the Centre for the Advanced Study of Collective Behaviour, funded by the Deutsche Forschungsgemeinschaft (DFG) under Germany's Excellence Strategy (EXC 2117-422037984). (https://www.dfg.de/en/research-funding/funding-initiative/excellence-strategy). M.C. and L.M.A. were supported by the Swiss State Secretariat for Education, Research and Innovation (SERI) under contract number MB22.00056 (https://www.sbfi.admin.ch/sbfi/en/home.html). L.M.A. was funded by a Max Planck Research Group Leader Fellowship (https://www.mpg.de/career/max-planck-research-groups). M.C. and G.A.N. received funding from the International Max Planck Research School for Quantitative Biology, Ecology and Evolution (https://imprs-qbee.mpg.de/). The funders had no role in study design, data collection and analysis, decision to publish, or preparation of the manuscript.

**Competing interests:** The authors have declared that no competing interests exist.

transect. Spatial variability can be observable (e.g., a change in habitat cues), but also might be unobservable, such as subtle changes in the rewards earned from familiar behavior. Exposure to observable variability may cause neurological changes that trigger learning [1], leading animals to explore the new environment to identify novel resources or risks [2,3]. Yet, trial and error exploration and learning in new environments may be extremely costly. For example, toxicity risks of novel food items may be largely unknown to the individual [4]. Longstanding theory has thus predicted that it should be advantageous to preferentially rely on social information as a less costly alternative to individual learning [5,6]. However, as the proportion of social learners increases, they rely on an ever smaller set of individual learners who are actively sampling the environment, causing the average fitness of the population to fall—an outcome referred to as Rogers' paradox [7]. Furthermore, the social learning might also come at some cost, especially under resource scarcity, either as an opportunity cost for time spent observing rather than foraging, or as increased likelihood of aggressive interactions due to resource defense [8].

Thus, individuals should not always socially learn, and an abundance of theoretical work supports the evolution of selective, flexible learning strategies [6,9–14]. Strategies should evolve that allow individuals to alternate between the acquisition and exploitation of socially learned behavior, at least until further variability is, or is expected to be, encountered [11,15]. Additionally, social learners should not necessarily copy all information equally, but rather strategically decide on who or what to copy [14]. Selective reliance on social learning in uncertain conditions has been identified by various experiments across taxa (see Kendal and colleagues [16] for review), although environments or rewards have rarely been deliberately manipulated.

One particularly uncertain context for wild animals is after migration or dispersal, as they are exposed to new physical and social environments. Several key studies give insight into how immigrants use social information. Migrant pied flycatchers (*Ficedula hypoleuca*) were shown to benefit from social information gleaned from resident great tits in the context of nest site selection [17,18]. Newly immigrated meerkats (*Suricata suricatta*) were found to adopt the wake-up time of residents of their new social group [19]. In chimpanzees (*Pan troglodytes*), female immigrants adopted residents' nut-cracking tradition, even if it was less effective than their own preferred tradition [20]. However, captive chimpanzees were shown not to adopt resident dietary preferences after migration [21]. Migrant male orangutans (*Pongo abelii*) were found to preferentially peer at local residents [22]. In vervet monkeys (*Chlorocebus pygerythrus*), males (the dispersing sex) were shown to preferentially copy dominant females who form the stable core of their social system [23]. When rewards were varied, males employed a pay-off biased learning strategy that overrode their model bias, while philopatric females continued to prefer to learn from other dominant females [24].

Formal models and experiments have addressed why immigrants should strategically use social learning after moving to a new environment. Firstly, under spatial variability, migration can introduce nonadaptive behavior that favors strategic learning that acts as a filter for both immigrants and residents [25]. This filtering is not as useful under temporal variability, as the entire population might possess nonadaptive behavior after a temporal change. Furthermore, migration results in experience-structured populations where residents would be preferred demonstrators because they might have gained more adaptive information as a result of longer experience in a particular environment [26]. A recent virtual foraging experiment on humans supported this difference between types of variability, showing that immigration across spatially variable environments temporarily increased reliance on social information more-so than temporal changes in rewards [26].

Great tits' social system and learning abilities have made them a proven study system for questions linking social learning to social network structure and dynamics. During the winter

months, great tits adopt a fission–fusion social system whereby birds join and leave mixed-species foraging flocks over the course of the day [27]. Tits are more socially tolerant compared with the spring breeding season, and belonging to winter flocks enhances their foraging abilities, either for patch discovery or the social transmission of novel foraging behavior [28–31]. Prior empirical studies have shown that naive great tits can acquire foraging behaviors with a conformist transmission bias, meaning that they disproportionately learned the most frequently observed option [29,32–34]. A follow up study found that this conformist bias did not create a trap in temporally variable environments where the rewards of options abruptly changed [32]. Tits adjusted their behavior to this change by relying mostly on individual learning, but retained a frequency-dependent conformity production bias [32]. However, the effect of spatial variability on tits' social learning has not yet been tested, even though spatial variability is hypothesized to drive the evolution of social learning strategies more-so than temporal variability [25] and could be more likely to trigger adaptive changes in learning strategies.

To test the effect of spatial variability on social information use, we conducted a cultural diffusion experiment in captivity using 18 micro-populations, each containing 8 wild-caught great tits ($N = 144$ birds). Each population was provided with an automated puzzle box with 2 possible solutions that gave access to either the same, or differing reward, and one "tutor" individual that was trained to use one solution. After this "seed" solution had spread and established as a behavioral tradition, we simulated immigration events between pairs of populations with opposing established traditions. For example, immigrants from a source population that preferred pushing the puzzle door right would be moved to a destination population that preferred left.

Each pair of source and destination populations was in 1 of 4 conditions, defined by a $2 \times 2$ factorial design. We manipulated environmental cues: populations had either symmetric, identical environments ($E_s$ condition) or asymmetric environments ($E_a$ condition) mimicking their natural habitat of pine or deciduous forests. We also manipulated puzzle payoffs, where puzzle solutions offered either symmetric or asymmetric payoffs (Fig 1B). In the symmetric payoff condition ($P_s$), both solutions gave access to a high-value mealworm reward before and after immigration. In the asymmetric payoff condition ($P_a$), the tutor's solution gave access to medium-value buffalo worms, while the alternative solution gave access to low-value sunflower seeds. After the immigration event, the payoffs changed such that the immigrants' preferred solution would still give access to buffalo worms, but the resident's preferred solution would give access to mealworms. Thus, immigrants would either need to use social information, or use individual learning by sampling themselves to discover that changing preferences would yield a better payoff. We note that seeded traditions were balanced across the design to avoid any confound with a bias for a particular solution (which we did not find). Finally, our design meant that immigrants would change both physical and social environments, while residents would only change social environments. While this study is largely focused on the behavior of immigrants, residents provided valuable insight into behavioral changes caused by a change in social environment alone.

We hypothesized that immigrants would be most likely to adopt the resident solution using social learning under maximal spatial variability ($E_a,P_a$; see Fig 1C). We expected immigrants to adopt the resident solution more-so by individual learning in the $E_s,P_a$ condition, as there would be no change in habitat cues to indicate a need to alter their behavior (comparable with the purely temporal variability tested in Aplin, Sheldon, and McElreath [32]). We expected that immigrants might be somewhat influenced by residents in the $E_a,P_s$ condition given the change in environmental cues, although there was no incentive to completely adopt the resident preference given that it provided the same reward. We expected immigrants would be least likely to switch preferences under minimal spatial variability ($E_s,P_s$). If social variability

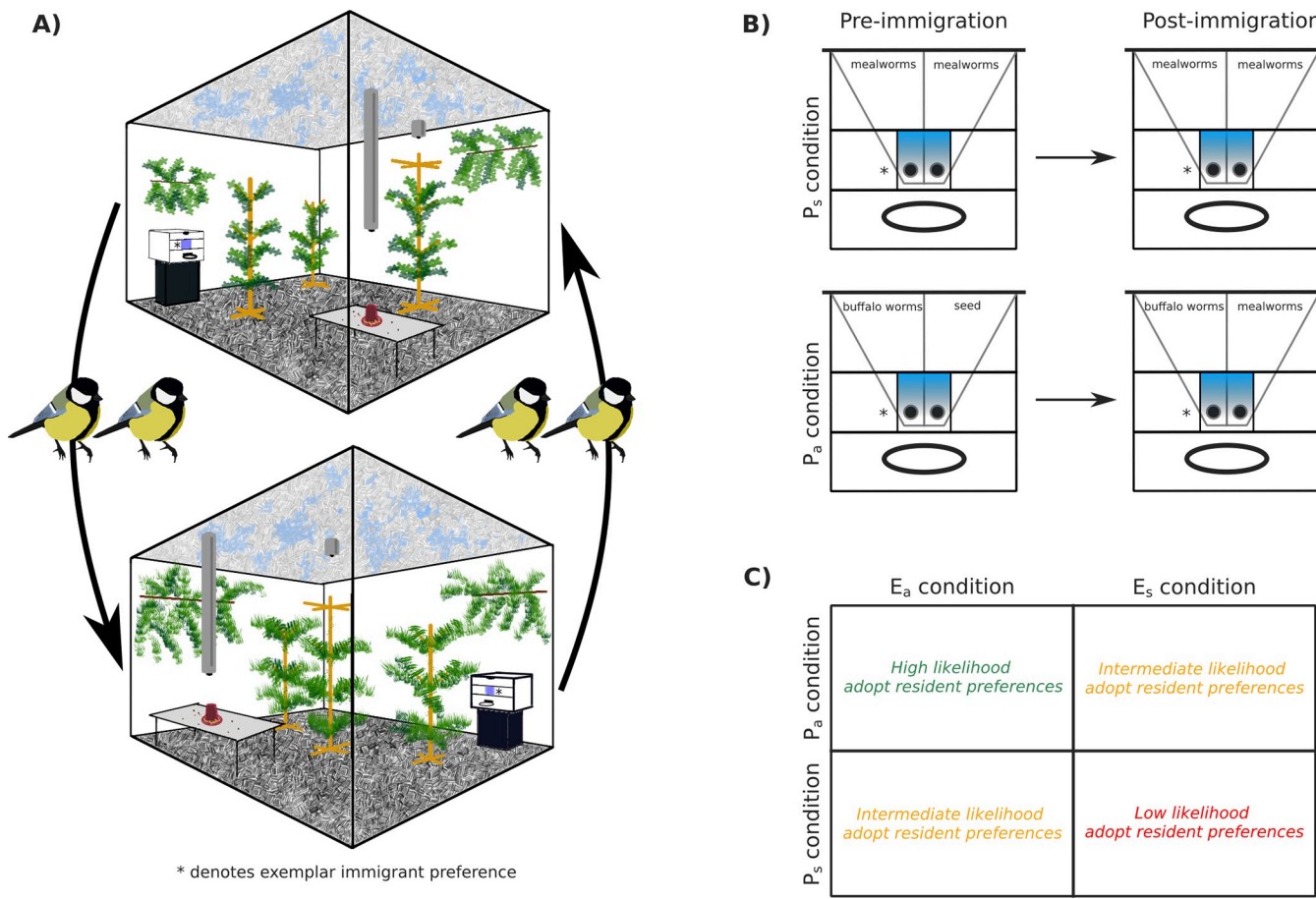

**Fig 1. Diagrams of experimental design and hypothesis.** (A) An example of immigration between 2 populations in the asymmetric environment condition ($E_a$), where 2 birds were swapped between populations with different habitat cues: deciduous (top) and pine (bottom) with opposing puzzle solutions. Prior to immigration, immigrants socially learned a seeded solution, demonstrated by the tutor (denoted by *). Seeded solutions were balanced across conditions. (B) Illustration of symmetric ($P_s$) and asymmetric ($P_a$) payoff conditions. In $P_a$ prior to immigration, the seeded solution was rewarded with buffalo worms, while the alternative solution was rewarded with sunflower seeds. After immigration, immigrants' preferred solution still gave access to buffalo worms, but the alternative resident solution gave access to the more preferred mealworms. (C) Diagram of our expectations about immigrant preferences under each condition in the 2 × 2 design. We expected that immigrants in $E_a,P_a$ (representing the maximal spatial variability) would be most likely to adopt the resident solution after immigration. The data underlying this figure can be found in our data and code repository (https://doi.org/10.17617/3.FXC12W).

alone triggered social learning, we expected residents to be more likely to be influenced by immigrants in $P_s$ conditions, since the mealworm reward would be more appealing that the buffalo worm reward obtained from the immigrant preference in $P_a$ conditions.

To test these hypotheses, we recorded the solving behaviors produced by birds before and after immigration to determine whether immigrants adopted the resident behavior of their new population. We then used Bayesian dynamic learning models that could differentiate between social and individual learning to analyze the learning mechanisms responsible for changes to immigrant preferences.

## Results

Before the experiment, we conducted pairwise food preference tests on a subset of birds ($N = 21$) to assess the relative value of the three food items used in our puzzles (sunflower seeds, buffalo worms, and mealworms). After 3 exposures to each food pair, birds chose buffalo worms over seeds roughly 75% of the time, and chose mealworms over buffalo worms

roughly 93% of the time. A Bayesian inverse reinforcement learning model estimated that birds valued mealworms (estimated value Q = 3.66) over buffalo worms (Q = 1.72) and buffalo worms over seeds (Q = 0.35). Food preference results are summarized in Table A in S1 Text and S1 Fig.

We then introduced 1 automated puzzle box to each population. The solving behavior spread in 17 of 18 populations, with a median of 5 (range: 3–7) out of 7 non-tutor birds recorded as solving the puzzle in each population before the immigration event (total 78 birds). Eight of these birds were recorded as having fewer than 5 recorded solves and were conservatively excluded from further analysis. The median latency to learn to solve the puzzle was 7.5 (range 1–31) days of exposure. In total, all solvers produced 200,109 recorded solutions, and non-tutor birds accounted for 145,426 of these. Solving frequency varied between birds, with non-tutor solvers producing a median of 1,128 solutions (range: 5–8,677) over the course of the experiment. Tutors produced solutions about 5 times more frequently as non-tutors (tutors: median 126 solutions per day; non-tutors: 22.8).

Prior to the immigration event, non-tutor birds overwhelmingly preferred the solution demonstrated by the tutor, with 97.3% of non-tutor solutions being that of their tutor's. This preference was not significantly different between conditions, age and sex classes, or between birds that went on to become immigrants and residents (Table B in S1 Text). Interestingly, but as expected, preference was estimated to be 3% lower in the symmetric payoff condition where both sides were equally rewarded with mealworms. Evidence that the behavior was socially learned was given by (1) the strong preference for the seeded solution in all populations that acquired the behavior; and (2) the 4 populations where the initial tutor did not solve also failed to innovate solving behavior and had to be supplied with tutors from other populations, or in the case of 1 more population, entirely excluded from the experiment.

## Behavioral preferences after immigration

We then swapped 2 solving birds between each pair of populations to simulate an immigration event. Immigrants did not appear to exhibit stress responses to the immigration event, and there was no noticeable increase in antagonistic interactions towards them. Immigrants continued using the puzzle box without interruption after immigration with a median delay of 0 days before starting to solve again. After the immigration, there were a median of 5 (range: 3–6) active solvers in each population, and a total of 33 immigrants were recorded as solvers ($E_a,P_a$ = 9; $E_s,P_a$ = 5; $E_a,P_s$ = 11; $E_s,P_s$ = 8), and 29 of the 33 immigrants produced at least 1 resident side solution within the 14 days after immigration. Immigrants produced their first resident-side solution after a median of 2 solutions. All 4 immigrants that did not produce the resident side were in $E_a,P_s$ condition in 4 separate populations.

Our hypothesis was generally supported when comparing the choices immediately preceding and following immigration (visualized in Fig 2), and over the longer term as the experiment progressed (Fig 3). Over the entire 14-day period following immigration, immigrants in the $E_a,P_a$ condition produced the largest proportion of resident solutions (proportion per condition compared to the other conditions: $E_a,P_a$ = 0.849, $E_s,P_a$ = 0.676, $E_a,P_s$ = 0.299, $E_s,P_s$ = 0.014). This effect was observed whether we took the overall proportion of solutions (individual immigrants thus weighted by solution frequency), or allowed each solver to be weighted equally and took the mean (individual's mean proportion (SD) per condition: $E_a,P_a$ = 0.891 (0.129), $E_s,P_a$ = 0.835 (0.235), $E_a,P_s$ = 0.269 (0.422), $E_s,P_s$ = 0.208 (0.369)). Immigrants in the $E_s,P_a$ condition also eventually adopted the resident solution, but with a delay relative to the $E_a,P_a$ condition (Figs 2 and 3). This delay hinted at a difference in learning mechanism, which we return to in the following selection.

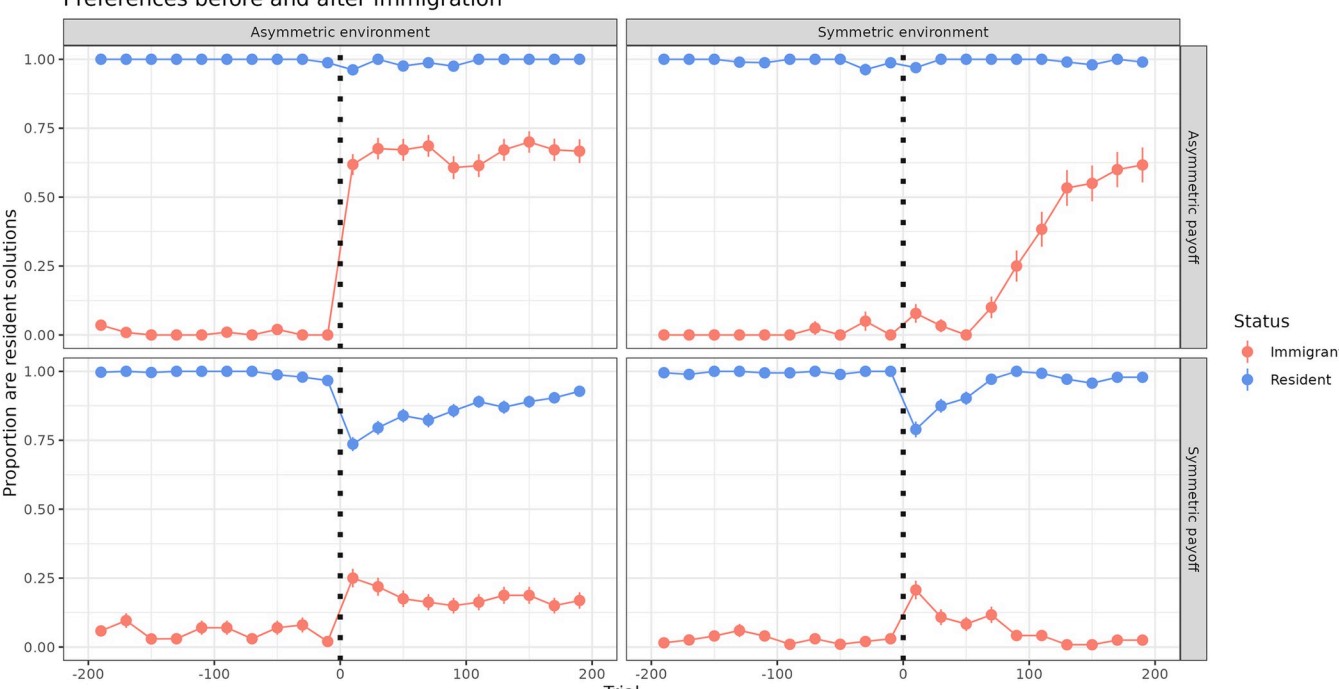

**Fig 2. Behavioral preferences immediately before and after immigration.** Mean, 95% CI of solutions produced by immigrants (blue) and residents (red) that were the resident preference in the destination population. Subset to the 200 choices before and after the simulated immigration event (dotted line). Prior to immigration, both immigrants and residents preferred the seeded solution in their respective populations. Following immigration, immigrants were most likely to adopt the resident solution when payoffs were asymmetric ($P_a$ conditions). They adopted the resident solution fastest under $E_a,P_a$. Immigrants were least likely to adopt resident solutions under $E_s,P_s$, although more immigrants adopted the resident solution in $E_a,P_s$. The data underlying this figure can be found in our data and code repository (https://doi.org/10.17617/3.FXC12W).

We also found interesting perturbations in the preferences of immigrants and residents in the $P_s$ conditions where both immigrants and residents appeared to be influenced by each other in the short term. Two immigrants came to prefer the resident preference (Fig 3, $E_a,P_s$ condition), and surprisingly, 9 residents used the immigrant preference more than 50% of the time (S2 Fig, $P_s$ conditions). Two of these had learned the puzzle after immigration, potentially socially learning from observations of immigrants. The commonality between the remaining 7 were that they were very low-frequency solvers (range: 6–89 total solves) compared to the median (1,128 total solves). We infer from this pattern that social variability can increase reliance on social information, as birds generally did not sample the opposing option prior to immigration. However, social influence was not generally strong enough to overcome the value of their original preference, gained through previous experience.

In order to understand how preferences changed over time, we conducted a logistic GLMM summarized in Table C in S1 Text. On the first day after immigration, immigrants were most likely to produce the resident solution in the $E_a,P_a$ condition (estimated $P = 0.896$). Immigrants in the $E_s,P_a$ condition were significantly less likely to do so, with an estimated $P = 0.50$. Even less likely was the $E_a,P_s$ condition at $P = 0.012$ and $E_s,P_s$ condition at $P = 0.069$, although this was not estimated to be significantly different from $E_a,P_s$ condition.

There was considerable variation in the solving experience of immigrants prior to the immigration event (median (Q1, Q3): 180 (7, 1346) solutions), caused by variable solving rates (range: 1 to 260 solutions per day) or the latency to have learned to solve the puzzle. The amount of this prior experience negatively influenced the probability of producing the resident

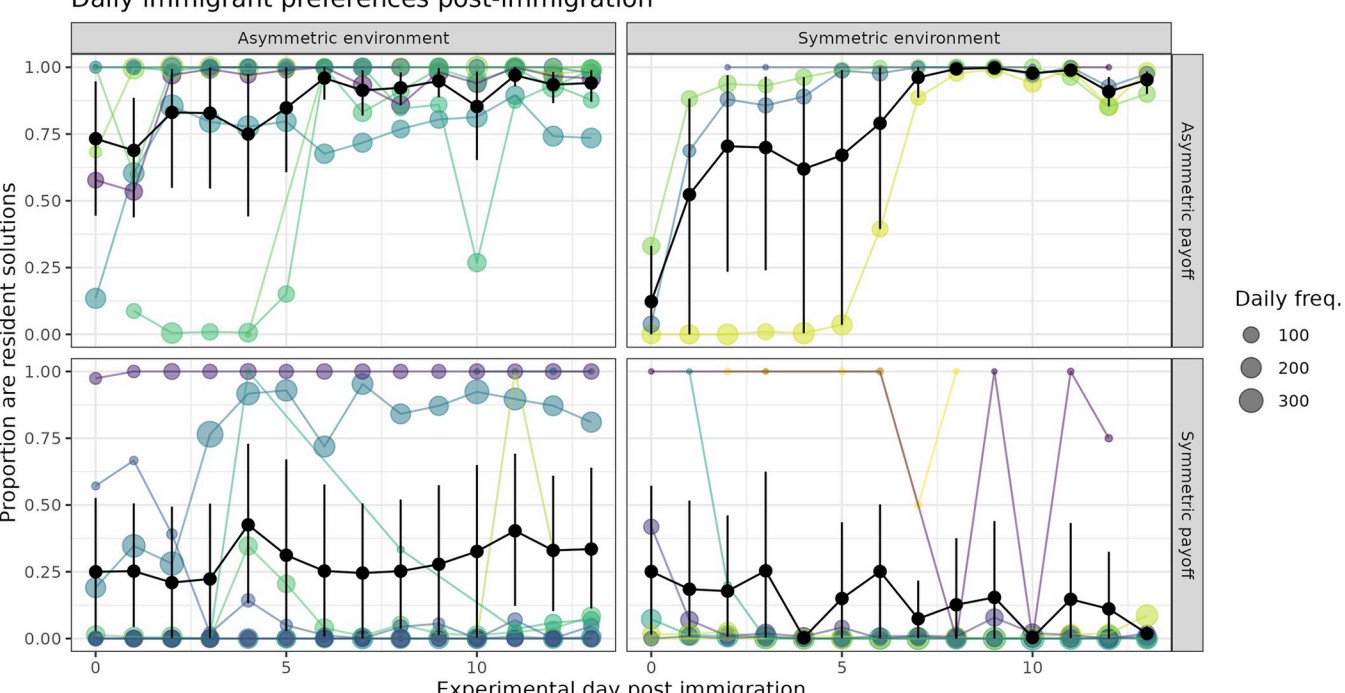

**Fig 3. Daily preferences of immigrants after immigration.** Proportion of immigrant solutions which were resident solutions (y-axis) over experimental day (x-axis). Colored lines are individual immigrants, with the daily solving frequency indicated (size). Black lines indicate mean, 95% CI. Immigrants were most likely to adopt the resident solution when payoffs were asymmetric. They adopted the resident solution faster when the environment was asymmetric, with a high adoption rate on day 0, compared to the symmetric environment condition. Immigrants were least likely to adopt when payoffs and environment were symmetric, although more immigrants adopted the resident solution when the environment was asymmetric. The data underlying this figure can be found in our data and code repository (https://doi.org/10.17617/3.FXC12W).

solution on the first day after immigration, although this did not meet our criterion for significance of $p < 0.05$ (GLMM; normalized experience—$\beta = -1.040$, $SE = 0.674$, $p = 0.124$). The interaction between experimental day and condition revealed that the probability of producing a resident solution significantly increased with each passing day in the both payoff asymmetric conditions (GLMM; day $\beta = 0.223$, $SE = 0.008$, $p < 0.001$, day: $E_s$ condition—$\beta = 0.431$, $SE = 0.023$, $p < 0.001$). It significantly decreased in both environment symmetric conditions, more-so when the payoffs were also symmetric (GLMM; day: $P_s$ condition—$\beta = -0.032$, $SE = 0.013$, $p = 0.015$, day: $E_s$ condition: $P_s$ condition—$\beta = -0.765$, $SE = 0.034$, $p < 0.001$). Age and sex did not significantly affect the probability of producing the resident solution and were not included in the final model.

## Learning models

Immigrants who began to prefer the resident-side solution could either have been influenced by social information provided by the residents, or could have independently decided to switch due to personal experience with the other solution. In order to better understand these potential mechanisms, we fit a set of 4 dynamic learning models, described in Materials and methods, to the first 200 choices of all individuals after immigration. We were especially interested in the time-period immediately after immigration, as preferences stabilized after several days (Fig 3), and the acute effect that immigration might have on learning might dissipate as immigrants adjusted to the new environment. These models estimated learning parameters in each condition for both residents and immigrants (summarized in Tables D–G in S1 Text).

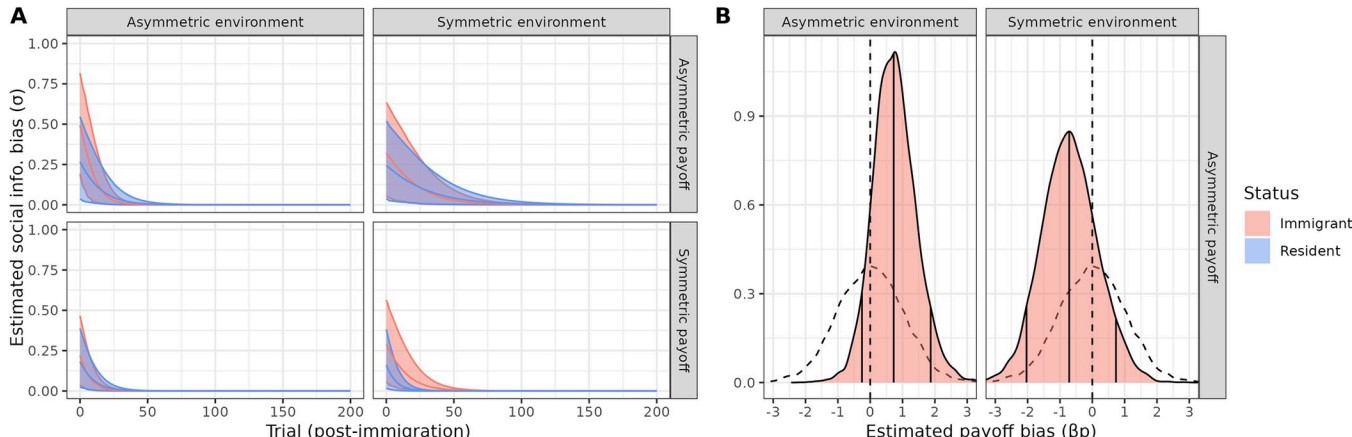

**Fig 4. Learning parameter estimates related to social information for best fitting model SL3.** (A) Posterior distributions (mean, 89% HPDI) of social information bias ($\sigma$) over time since immigration. Individuals who immigrated to a different environment with different asymmetric payoffs were most immediately influenced by social information. (B) Posterior distributions of estimated payoff bias ($\beta_p$) for conditions with asymmetric payoffs. Mean, 89% HPDI indicated by vertical lines. When birds immigrated to a novel environment, they strongly weighted observations of the higher-payoff solution. The data underlying this figure can be found in our data and code repository (https://doi.org/10.17617/3.FXC12W).

We found that all models which allowed individuals to be influenced by social information, rather than personal experience alone, predicted the data significantly better as measured by WAIC score, summarized in S3 Fig. The lowest WAIC score was obtained by model SL3, which estimated a slope for the social influence parameter, a payoff bias and new associate bias parameters for social information (summarized in Fig 4, individual birds' point estimates in S4 Fig). In all conditions, both residents and immigrants were estimated to be influenced by social information (although immigrants more-so), and this influence was estimated to decline over time in all conditions (Fig 4A). Importantly, the $E_a,P_a$ condition was estimated to have the highest social influence immediately following immigration ($\sigma$(89% HPDI) = 0.49(0.18, 0.81), Fig 4A), and immigrants were estimated to be disproportionately influenced by observations of the higher payoff solution ($\beta_p$ = 0.75(−0.26, 1.83), Fig 4B). When immigrants were socially influenced in the $E_s,P_a$ condition, they were estimated to be more influenced by observations of the lower payoff solution by fellow immigrants (Fig 4B). Their eventual shift towards the residents' preference was thus more-so driven by individual learning rather than social influence from residents.

Given that the payoff bias parameter's formalization required options with different payoffs, it was not informative for symmetric payoff conditions. However, SL3 also fit a new associate bias ($\beta_n$) to estimate whether immigrants and residents were more influenced by their new associates. We found a trend that residents in both symmetric payoff conditions were influenced by immigrant solutions (residents in $E_a,P_s$: $\beta_n$ = 1.03(−0.03, 1.95), $E_s,P_s$: $\beta_n$ = 0.93(−0.67, 2.45)), whereas posterior $\beta_n$ distributions were centered around 0 in the $P_a$ conditions, suggesting that residents were assessing the rewards and were not influenced by the lower-rewarded option (S5 Fig). This bias explained the perturbation in residents preferences following immigration in Fig 2. Finally, we fit an ancillary model that estimated a tutor bias, but found no evidence for the disproportionate influence of tutors on immigrants (S6 Fig).

When taken together with the general pattern in Fig 2, these mechanisms point to a clear behavioral difference between the $E_a,P_a$ and $E_s,P_a$ conditions: immigrants rapidly changed their preferences using social information when experiencing asymmetric environments and were disproportionately influenced by the social observation of asymmetric payoffs. Meanwhile, when environments were symmetric, immigrants were less influenced by social

information, and they did not disproportionately attend to the higher payoff solutions. Individual learning was the most likely mechanism behind their adoption of the resident preference.

## Discussion

By simulating immigration events for individuals between groups with different traditions in different environments or different foraging pay-offs, we could disentangle flexibility in learning strategies associated with these 2 sources of variability. Our design also allowed us to test birds' responses to social variability alone. We found that immigrants were most likely to adopt the residents' preference when both environments and foraging payoffs differed, and relied more on social learning, likely with a pay-off bias. When environmental cues remained the same and only the payoff landscape changed, immigrants began to prefer the resident solution after a slight delay, more likely shifting due to individual learning. Immigrants were less likely to adopt the residents' preference under both symmetric payoff conditions, and least likely when environments and payoffs were symmetric. By contrast, residents, who only changed social partners and not locations, essentially never changed their preferences in the asymmetric payoff conditions, but did initially sample the immigrants' preference in the symmetric payoff conditions. These data suggest that residents were also able to perceive the rewards that others received at the puzzle box, and that these observations initially influenced their choices immediately following immigration. Our results therefore generally supported our hypothesis with evidence for a stronger effect of spatial variability on behavioral responses.

The results of our experiment demonstrate that, in tits, social learning is not inflexible or unbiased, but rather can exhibit adaptive flexibility in response to environmental change or uncertainty [5,12–14,16,35,36]. Indeed, for over a century it has been recognized that phenotypic plasticity, or the expression of different phenotypes depending on environment, is crucial for success in adapting to novel environments [37,38]. Our current study points to specific plasticity in learning strategy to changes in the environment. Our learning model analysis indicates the cognitive mechanism responsible for immigrants' rapid adoption of resident preferences in the $E_a,P_a$ condition was the activation of a learning strategy where the rewards of others were accounted for when making future decisions. By inducing immigration events in controlled conditions, we further show that these changes in learning strategies, at least in great tits, are not due to underlying life-history state, e.g., tied to hormonal changes during dispersal [22]. Rather, they represent flexible shifts triggered by experience of new environments. However, we do not discount the role that hormonal changes may play in other contexts or species and suggest that both mechanisms might operate in an additive effect.

In primates, immigrant individuals have been shown to conform to their new group's preferences [20], although this depended on behavioral context [21]. Specific reliance on a pay-off biased strategies by immigrants has been demonstrated in captive chimpanzees [39], and in male vervet monkeys, which are the dispersing sex [24]. The results of our experiment add to these findings, yet suggest that tits do not always adopt the preferences of their new group, but rather that this depends both on whether there is spatial variability in payoffs (in the sense that rewards of options has changed with physical location) and environmental cues. When uncertainty was high, either in the beginning of the experiment when birds were completely naive to the puzzle, or after immigration when payoffs and environmental cues changed, the tits relied on social information with a conformist bias in the first case (similar to initial acquisition in previous studies [29,32]), and a payoff bias in the second case. Immigrants' retention of preferences in both $P_s$ conditions, while conforming to theory, disagreed with anecdotal evidence from a limited number of birds suggesting that knowledgeable migrant birds adopt local

preferences under symmetric payoffs [29]. However, unlike primates, tits likely do not use foraging behavior as a social identifier, and do not need to conform to mitigate potential aggression [20]. Tits do not appear to conform for the sake of conformity, but rather need a good reason (i.e., better reward) to adopt resident preferences once having already learned how to use the puzzle.

Theory has suggested that payoff-biased learning should be favored in temporally varying environments [14] and evolves in spatially varying environments only if its cost is equal to or less than the cost of a conformist strategy [25]. We would suggest that a payoff bias is less costly than a conformist strategy for great tits, especially given their winter fission–fusion social system. Firstly, it may be difficult for immigrants to assess which of their group members are residents, as opposed to other immigrants from a third location. Secondly, the number of residents may be less than the number of immigrants, leading to a case where the frequencies of behaviors are not indicative of their adaptive value. This was also the case in our experiment, where residents varied substantially in their solving rate, leading to some replicates where the large majority of behavioral information was only available from one resident. Thirdly, we argue that payoff assessment is a cognitively straight-forward task in our design, since immigrant birds are already familiar with both buffalo worms and mealworms, and these food items are visually distinct with mealworms being about twice as large as buffalo worms. Birds carry the food items away from the puzzle in their bill before processing and eating them. Immigrants only had to attend to others' solutions and recognize the reward earned by the choice of a conspecific was a food item that they preferred more-so than the reward they had been earning from their own behavior. In open fission–fusion systems, the observation of payoff cues is therefore likely a more reliable means of identifying adaptive behavior. Thus, this is a case where theory and natural ecology of a species match to experiment.

Our experiment also highlights the distinction between transmission biases versus production biases. The birds in our study strongly preferred their tutor's seeded solution, with results in line with prior work on tits [29,32]) that showed that great tits socially learn and acquire novel behaviors with a conformist transmission bias. The pay-off bias that we evidence after immigration is a bias in social influence, which acts on production, rather than a transmission bias (as emphasized by Chimento and colleagues [40]). Thus, while the prediction presented by Nakahashi and colleagues [25] that conformist transmission bias out-competes payoff transmission bias in spatially varying environments may be valid, it might not necessarily speak to the evolution of payoff-biased social influence on production, as would be the case for immigrants that held previous knowledge.

In most previous studies that rely on observations of natural immigration events [29,41], it is impossible to distinguish between changes in social environments and changes in physical environments. In our study, changes in social environment alone did not trigger a payoff bias. This is perhaps to be expected, given that the tits exhibit fission–fusion-foraging flocks during winter when their reliance on social information is highest [42]. Tits' associates may change within and between days, even though their home-range may only less than a square kilometer [27,43], and likely not too spatially variable. However, tits can move several kilometers, and occasionally irrupt in response to environmental triggers, traveling hundreds of kilometers to take up residence in a novel environment [44]. Therefore, it is plausible that environmental variability, rather than social variability, would trigger payoff-biased social learning. A prior study on tit's response to purely temporal changes in payoffs found that birds relied more-so on individual learning rather than social information, with evidence for an ongoing conformist bias, but no evidence for a payoff-bias [32]. This finding is consistent with the results from our experiment in the $E_s,P_a$ condition, where immigrants preferred the resident solution after a slight delay.

In summary, our results have shown that birds change their behavior most rapidly when immigrating to a novel environment, with behavior shifting to match the resident preferences. We further showed that this behavioral change was achieved through the use of payoff-biased social information. Our study therefore adds empirical support for the hypothesis that spatial variability can drive the evolution of flexible social learning strategies. Our results further suggest that, in tits, spatial variability is a stronger trigger for adaptive shifts in learning biases than temporal variability. Finally, we highlight that social learning strategies that act on production may not necessarily be the same as strategies that act on acquisition.

## Materials and methods

Our experiment consisted of 18 micro-populations of wild-caught great tits ($N$ = 144 birds), kept in outdoor aviaries of $3 \times 4 \times 3$ m volume, and given ab libitum access to water, a seed-based food mix, and a puzzle box containing live food. Each population began with 8 birds, one of which was a "tutor" who had been previously trained to solve an automated puzzle box using the training method described in Chimento, Alarcón-Nieto, and Aplin [33]. We balanced age and sex (77 males) to the best of our ability given the stochastic sampling of catching wild birds. Populations contained a mix of first-year and adult individuals (69 adults) and were caught from various sites within 10 km of the Max Planck Institute of Animal Behavior (47.768064, 8.996331). The puzzle box consisted of a sliding door that could be pushed left or right. Pushing in either direction was equally difficult, but the reward was side-specific: either mealworms, buffalo worms, or sunflower-seed, depending on the experimental conditions described below. Puzzle boxes were automated and used an RFID antenna to record identities of birds, the time at which they landed on the perch, the side which they pushed the door open (recorded by infrared sensors), and the time at which they left the perch [29,33,45]. From this data, we could track the behavioral productions of each solving bird over the course of the experiment. Puzzle boxes were placed in a corner facing the center of the room, allowing all birds in the room to potentially observe it.

The experiment took place over 2 time periods, from January to March in 2021 ($N$ = 48 birds) and 2022 ($N$ = 88 birds). The experiments were conducted during winter when tits are most tolerant of other birds and form foraging flocks. This is outside the main dispersal times in tits, which tend to occur during autumn or early spring, but flocks can roam extensively outside individuals' core home-range throughout winter [27]. The experiment began with a diffusion period lasting between 15 and 21 days to allow for the solving behavior to spread. Once the solving behavior was learned by 3 or more birds, the populations underwent an immigration event, where 2 non-tutor solvers from each population were exchanged with 2 non-tutor solvers from another population (henceforth immigrants). This was done in the night by moving nest boxes, such that birds went to sleep in their nest box in their source population, and woke up in their nest box in their destination population.

Importantly, immigration occurred between populations with opposing resident solutions. We attempted to balance immigrants based on sex (19 male), age (13 adult), and solving rate (17 with >100 solutions prior to immigration). Two more non-solving birds were removed from each population at this point for ethical considerations, as non-solving birds were considered to be unlikely to learn the task. The experiment continued for 14 days following the immigration event. After the experiment ended, the birds were released back into the wild at the location where they were caught.

Populations were split into 4 conditions, in a $2 \times 2$ factorial design where we manipulated environmental conditions and payoff conditions. Aviaries contained foliage from one of 2 different environments that great tits are naturally found in: pine or deciduous forest.

Additionally, the puzzle box doors were also either blue or gray, depending on the environment type. In the symmetric environment ($E_s$) condition, immigrants moved to a population with an identical environment to their original population. In the asymmetric environment condition ($E_a$), immigrants moved to a population with a different environment.

In the symmetric payoff ($P_s$) condition, pushing from either side of the puzzle box door gave access to a reservoir of mealworms, and the same was true after the immigration event. In the asymmetric payoff condition ($P_a$), populations began with a puzzle box that gave either buffalo worms (the resident solution that the tutor had been trained on), or sunflower seeds (see Fig 1B). Buffalo worms are smaller than mealworms and less preferred by the tits (see section Food item preference tests). Both buffalo worms and mealworms are preferred to sunflower seeds [32]. After the immigration event, the food rewards were changed such that the resident solution gave access to mealworms, while the immigrant solution gave access to buffalo worms. Therefore, immigrants would not necessarily realize that the payoff structure had changed without using social information or sampling the other solution themselves.

Four of the $E_a,P_s$ replicates were performed in 2021, with 2 more in 2022 to test for year effects. Two of the $E_s,P_s$ replicates were performed in 2021, and 2 more were performed in 2022. All of the $E_s,P_a$ and $E_a,P_a$ replicates were performed in 2022. Altogether, $E_s,P_s$ and $E_a,P_a$ had 4 replicate populations, $E_s,P_a$ had 3 replicates (due to the exclusion of 1 population where the solving behavior did not spread), while $E_a,P_s$ had 6. We also note that the tutors of 4 populations did not solve within the first 7 days and were replaced with tutors trained on the same solution from another population.

## Puzzle preference models

In order to assess non-tutor birds' preferences relative to the seeded solution in source populations prior to immigration, we conducted a GLMM using the lme4 and lmerTest packages in R [46–48] where the dependent variable was a birds' proportion of solutions that were the seeded solution introduced by the tutor bird predicted by the age, sex, and a three-way interaction between immigrant status, payoff and environment conditions. Year was included as a random effect. Data was subset to only days prior to the immigration event.

In order to analyze how non-tutor birds' preferences changed over time after the immigration event, we used a logistic GLMM where the dependent variable was whether a solution was the seeded solution in the destination population, predicted by normalized experience, and a three-way interaction between experimental day, environment and payoff conditions. ID nested within year was included as random effects. Data was subset to only days after the immigration event took place. Normalized experience was measured as the total number of solves minus the mean number of solves divided by its standard deviation. Experimental day 0 represented the day immediately following immigration. For both models, summary tables were created using the stargazer package [49].

## Learning models

In order to understand the learning mechanisms by which immigrant birds' changed or kept their preferences, we analyzed the first 200 choices made immediately following immigration using Bayesian reinforcement learning models. Birds solved at variable rates, but these choices were generally made within the first few days following immigration. We first fit a model that only allowed for individual learning of preferences, and then fit decision-biasing models, which estimate the degree to which social information influences preferences [10,26,32,50]. These models assumed that individuals keep a memory of social observations ($S$) that were considered along with an expected value ($Q$) calculated from personal experience to influence

choice probability. We compared models using the widely applicable information criterion (WAIC) scores.

**Individual learning model (IL).** The simplest model estimated 2 parameters: a learning rate ($\rho$) and a temperature parameter ($\alpha$). $\rho$ is constrained between 0 and 1 and was estimated on a logit scale. $\alpha$ is constrained to be above zero and was estimated on a log scale. When a bird produced a solution $k$ at the puzzle-box, its expected value for chosen option $k$, $Q_{k,t+1}$, was updated as a function of the reward ($\pi_k$) weighted by $\rho$:

$$Q_{k,t+1} = \rho\pi_k + (1 - \rho)Q_{k,t}. \tag{1}$$

The un-chosen option was updated with $\pi_k = 0$. Expected values were transformed into an individual choice probability $I$ in Eq 2 using a softmax function where repertoire $Z_i$ was a vector of both behaviors and temperature parameter $\alpha$ controlled sensitivity to differences in expected values:

$$I_{i,k,t} = \frac{\alpha exp(Q_{k,t})}{\sum_{k \in Z_i} \alpha exp(Q_{k,t})}. \tag{2}$$

When $alpha > 1$, individuals were more likely choose the option with a larger Q-value. When $\alpha = 1$, individuals were linearly sensitive to differences in Q values, and when $\alpha < 1$, they were less sensitive.

**Social learning model (SL1).** The first social learning model estimated a learning rate ($\rho$), a risk-appetite parameter ($\alpha$), and a social information bias ($\sigma$) with a frequency cue. $\sigma$ was estimated on a logit scale. Rather than relying on personal experience alone, socially observed solutions could bias individuals' final choice probabilities. Solutions $k$ observed by individual $i$ at time $t$ were accounted for by frequency cue $S$:

$$S_{i,k,t} = \frac{\sum_{t-15m}^{t} n_k}{\sum_{k \in Z_i} \left(\sum_{t-15m}^{t} n_k\right)}, \tag{3}$$

where the number of observations of solution $k$ was summed over a period of 15 min prior to $t$ and was normalized by the sum of all solutions observed. This social information was combined with personal information into a probability of choice $P_{i,k}$. The relative importance of social information was determined by $\sigma$. Here, a low value would mean agents were predominantly influenced by their personal experience, and a high value meant that they were predominantly influenced by their social observations.

$$P_{i,k,t+1} = (1 - \sigma)I_{i,k,t} + \sigma S_{i,k,t} \tag{4}$$

The calculation of $S$ required us to define which solutions could be observed by a focal bird. Given the position of the puzzle box within the aviaries, all birds could potentially observe it, as well the rewards that birds carried away in their bills. We assumed that if solution $k$ was produced by any other birds in the 15 min prior to the current solution, it was observed; 10 min and 20 min were also tested, but resulted in similar estimates with poorer fits.

**Social learning models with payoff and new associate biases (SL2).** In SL2, we implemented a payoff bias $\beta_p$ and a new associate bias $\beta_n$ in a log-linear term $L$:

$$L_k = exp(\beta_p \bar{\pi}_k + \beta_n a_k). \tag{5}$$

Payoff cue $\bar{\pi}_k$ was the average reward observed for choice $k$. Thus, $\beta_p$ estimated whether individuals were assessing the value of the food items obtained by others from either side of their puzzle, where $\beta_p = 0$ indicated that an individual was insensitive to observed payoffs, $\beta_p$

$< 0$ indicated that they would be more influenced by behaviors with relatively lower payoffs, and $\beta_p > 0$ indicated that they would be more influenced by behaviors with higher payoffs. Due to its formalization, this parameter could only be meaningfully estimated for asymmetric reward conditions. The new associate cue $a_k$ represented the proportion of observed solutions $k$ which had been performed by new associates. The log linear term was then included by modifying Eq 3:

$$S_{i,k,t} = \frac{\sum_{t-15m}^{t} n_k L_k}{\sum_{k \in Z_i}\left(\sum_{t-15m}^{t} n_k L_k\right)}. \tag{6}$$

**Social learning model with time-varying slopes (SL3).** We expected $\sigma$ to potentially weaken over time, as evidenced in a previous experiment on human immigration [26]. We modified SL3 to include a slope parameter $\beta_\sigma$, allowing us to estimate whether social influence increased or decreased with each trial $t$:

$$logit(\sigma_t) = \mu_\sigma + \beta_\sigma t. \tag{7}$$

The values of $\rho$, $\sigma$, $\beta_p$, $\beta_n$ and $\beta_\sigma$ were estimated within conditions $C$ and immigrant status $S$, with random effects of individuals (e.g., $logit(\rho) = \alpha_{[ID]} + \mu_{\rho,[C,S]}$). We did not have any specific predictions about differences in $\alpha$ between conditions, and this was estimated as a grand mean with random effects of individuals. For all parameters except $\sigma$, we assumed weakly informative normal priors for means (e.g., $\mu_\rho \sim N(0,1)$), making our model skeptical of large effects. For $\sigma$, we conservatively assumed a lower prior ($\mu_\sigma \sim N(-1,1)$) based on estimated $\sigma$ from a prior study [32]. We used non-centered varying effects with a Cholesky decomposition of the correlation matrix, following prior applications [26,50]. We ensured model convergence using Gelman's statistic $\hat{r} < 1.01$ and visually confirmed that chains were well-mixed by using rank histograms. We validated models with simulated data using known parameter values to confirm that models could accurately recover values. Code for simulation is available at the link in the data availability section. All models were fit with 5 chains and 5,000 iterations using R v. 4.0.2 [48] with Stan v. 2.27 [51] via Rstan v. 2.21.2 [52].

## Food item preference tests and payoff coding

To determine how birds perceived the relative rewards of the different food items, we performed preference tests prior to the diffusion phase of the experiment. In these trials, open bowls of sunflower, mealworms, and buffalo worms were provided in a table in the middle of 4 aviaries, and the choices by 21 individuals were recorded for 3 replicate trials in each aviary (summarized in S1 Fig). We found that birds preferred mealworms over buffalo worms approximately 93% of the time, while buffalo worms were preferred over seeds. We conducted a Bayesian inverse reinforcement learning model to estimate the values of each food item (results reported in S1 Text). We then used these point estimates to inform the rewards received in the learning models. We also used the values estimated by the inverse RL model to code birds initial knowledge state. Assigning $Q_{i,k,t=0} = 0$ for both solutions would be an unreasonable assumption that individuals have a 50% chance of producing either behavior immediately after immigration. All immigrants had a majority of their experience with the immigrant-side solution (0.97 of their solutions entire diffusion phase), and thus we coded $Q_{i,IM,t=0} = 1.72$, $Q_{i,RS,t=0} = 0$ in the asymmetric payoff condition, and $Q_{i,IM,t=0} = 3.66$, $Q_{i,RS,t=0} = 0$ in the symmetric payoff condition. This biased them towards their preferences upon immigration, yet assumed a nonzero probability that they might choose the alternative side.

### Ethics statement

All work was conducted by under a nature conservation permit and animal ethics permit from the Regierungsprasidium Freiburg, no.35-9185.81/G-20/100.

### Supporting information

**S1 Fig. Results from preference tests.** Proportion of choices (y-axis) by trial (x-axis), where each dot represents an individual bird. Birds preferred buffalo worms to seed, and mealworms to buffalo mealworms, and these preferences strengthened over time. The data underlying this figure can be found in our data and code repository (https://doi.org/10.17617/3.FXC12W). (PNG)

**S2 Fig. Daily preferences of residents after immigration.** Daily preferences of residents after immigration. Proportion of immigrant solutions which were resident solutions (y-axis) over experimental day (x-axis). Colored lines are individual immigrants, with the daily solving frequency indicated (size). A handful of residents in the $P_s$ conditions adopted the immigrant preference; however, they had either learned to use the puzzle after immigration, or had very low solving rates. The data underlying this figure can be found in our data and code repository (https://doi.org/10.17617/3.FXC12W). (PNG)

**S3 Fig. WAIC model comparison.** WAIC comparison of dynamic learning models. Black dots indicate out-of-sample deviance, "+" symbols indicate in-sample deviance. Gray bars indicate standard error, and black bars are the difference in standard errors. Vertical dotted line is the WAIC of the highest ranked model. Models with social learning components were better fit than individual learning alone, and the highest ranked model estimated payoff-biased social learning, a new associate bias, and a sloped change to the sensitivity to social information. The data underlying this figure can be found in our data and code repository (https://doi.org/10.17617/3.FXC12W). (PNG)

**S4 Fig. Individual point estimates for key parameters.** Point estimates of individual birds from Bayesian learning model SL3 (immigrant status as color). Point estimates shown for learning rate ($\rho$), social information bias ($\sigma$), payoff bias ($\beta_p$), new associate bias ($\beta_n$). The data underlying this figure can be found in our data and code repository (https://doi.org/10.17617/3.FXC12W). (PNG)

**S5 Fig. New associate bias estimates from model SL3.** Posterior distributions of an estimated new associate bias for residents and immigrants. Estimates above zero indicate that birds were disproportionately influenced by observations of solves by new associates. Estimates below zero indicate that individuals disproportionately ignored observations of solves by new associates. The data underlying this figure can be found in our data and code repository (https://doi.org/10.17617/3.FXC12W). (PNG)

**S6 Fig. Tutor bias estimates for immigrants.** We fit an additional model, similar to SL3, except that rather than information about whether a solution was produced by a new associate, we included information about whether a solution was produced by a tutor. This was done to determine whether immigrants were disproportionately influenced by the productions of tutors in their new populations. We found no evidence to support this in any condition. The data underlying this figure can be found in our data and code repository (https://doi.org/10.17617/3.FXC12W). (PNG)

**S1 Text. Supplementary Tables.** Table A. Relative value of food items. Table B. Non-tutor birds preferred the seeded solution. Table C. How did immigrant preferences change after the immigration event? Table D. Summary of model estimates, individual learning only (IL). Table E. Summary of model estimates, SL1. Table F. Summary of model estimates, SL2. Table G. Summary of model estimates, SL3.
(PDF)

## Acknowledgments

Authors thank Dr. Inge Müller, Dr. Daniel Zuñiga, and the MPI-AB animal care team for their support during this study.

## Author Contributions

**Conceptualization:** Michael Chimento, Lucy M. Aplin.

**Data curation:** Michael Chimento.

**Formal analysis:** Michael Chimento.

**Funding acquisition:** Lucy M. Aplin.

**Investigation:** Michael Chimento.

**Methodology:** Michael Chimento, Gustavo Alarcón-Nieto, Lucy M. Aplin.

**Project administration:** Michael Chimento.

**Resources:** Michael Chimento, Gustavo Alarcón-Nieto.

**Software:** Michael Chimento.

**Supervision:** Lucy M. Aplin.

**Validation:** Michael Chimento.

**Visualization:** Michael Chimento.

**Writing – original draft:** Michael Chimento.

**Writing – review & editing:** Michael Chimento, Gustavo Alarcón-Nieto, Lucy M. Aplin.

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
