## [Editor Report · Decision Letter 0]

31 May 2024

Dear Michael, 

Thank you for submitting your manuscript entitled "Immigrant birds use payoff biased social learning in spatially variable environments" for consideration as a Research Article by PLOS Biology.

Your manuscript has now been evaluated by the PLOS Biology editorial staff, as well as by an academic editor with relevant expertise, and I'm writing to let you know that we would like to send your submission out for external peer review.

Once your full submission is complete, your paper will undergo a series of checks in preparation for peer review. After your manuscript has passed the checks it will be sent out for review. To provide the metadata for your submission, please Login to Editorial Manager (https://www.editorialmanager.com/pbiology) within two working days, i.e. by Jun 04 2024 11:59PM.

Kind regards,

Roli

Roland Roberts, PhD

Senior Editor

PLOS Biology

rroberts@plos.org

---

## [Decision Letter · Decision Letter 1]

1 Aug 2024

Dear Michael,

Thank you for your patience while your manuscript "Immigrant birds use payoff biased social learning in spatially variable environments" went through peer-review at PLOS Biology. Your manuscript has now been evaluated by the PLOS Biology editors, an Academic Editor with relevant expertise, and by three independent reviewers.

You'll see that reviewer #1 says that your study is interesting, rigorous and compelling, but has a number of issues with the clarity of the manuscript (especially the Results and Fig 1). Reviewer #2 says this is a valuable contribution, and is well written. S/he simply had a few minor requests. Reviewer #3 is also positive, but wonders if the you are missing part of the story; I'm uncertain, but this may well stem from the same lack of clarity that was mentioned by reviewer #1.

IMPORTANT: After some discussion, we think that your paper would be best considered as a Short Report. As your manuscript is already concise and only has 4 Figures, no re-formatting is needed, but I've switched the article type in our system.

In light of the reviews, which you will find at the end of this email, we are pleased to offer you the opportunity to address the comments from the reviewers in a revision that we anticipate should not take you very long. We will then assess your revised manuscript and your response to the reviewers' comments with our Academic Editor aiming to avoid further rounds of peer-review, although might need to consult with the reviewers, depending on the nature of the revisions.

**IMPORTANT - SUBMITTING YOUR REVISION**

*Resubmission Checklist*

*Published Peer Review*

*PLOS Data Policy*

*Blot and Gel Data Policy*

Sincerely,

Roli

Roland Roberts, PhD

Senior Editor

PLOS Biology

rroberts@plos.org

REVIEWERS' COMMENTS:

Reviewer #1:

This is an interesting study using an experiment on captive great tits alongside Bayesian learning models to test how social information use by immigrant individuals is influenced by changes in habitat type and reward payoffs. The experiment is very cleverly designed and the analyses are rigorous and compelling, so I have only relatively minor comments on the clarity of the manuscript.

General comments:

My main comment is that, because of the way the paper is structured, with the Methods at the end, it is rather hard to understand the design and logic of the experiment. For instance, when you were discussing spatial variation in the introduction, I assumed that you were talking about the informational content of spatial cues, as this is central to what immigrants need to learn about. For instance, an immigrant may need to learn something like "in this new environment, the best caterpillars are found on oak trees". Figure 1 also gave me this impression, as it seemed from the pictures that the spatial location of cues (deciduous or pine) was associated with the reward distribution in a meaningful way (see specific comment below). It was only much later that I realised that when you referred to spatial variation you meant habitat differences - i.e. moving from one habitat type to another, but with no new informative spatial cues about where food might be found. Similarly, I did not immediately see the link between temporal variation and changes in payoffs - after all, if I am used to getting food rewards from the right-hand side but now the best rewards are on the left, that is a spatial, as well as a temporal change. I think providing a bit more detail on how the concepts covered in the introduction relate to the experimental design, and on how the experiment itself is structured, would really help readers to understand the study and reduce the need to flick back and forth between sections to make sense of it all.

There were also aspects of the results that I found hard to follow. In particular, at first I found it difficult to reconcile Fig 2, where immigrants in the Symmetric Environment, Asymmetric Payoff (Es, Pa) show a clear switch to the resident strategy, with later arguments about limited learning from residents when environments were symmetrical. The discussion later clarifies that this is because Es; Pa immigrants relied more on individual learning about the optimal solution, but this was not immediately obvious from the presentation of Results. Perhaps you could include a figure to illustrate how reliance on individual learning varies across conditions?

Specific comments:

Abstract and L82: State explicitly that experiments were in captivity. Given the authors' previous work, readers may assume (as I did) that the experiments were in the wild.

L56: It would be useful to clarify here why resource scarcity should generate costs of social information acquisition

L58: Typo - "should not copying"

L64-71: There are some relevant studies beyond primates that could be cited here - e.g. migratory flycatchers learning from resident great tits in Finland; immigrant meerkats adopting the waking-up times of their new groups.

L73: I suggest defining conformist transmission bias here for readers who may not be familiar with the formal definition.

L76: Why is spatial variation thought to be more important than temporal variation? A brief explanation would be helpful for context.

Figure 1: I found the figure a little hard to follow. In A, the pictures seem to indicate that environmental cues were spatially associated with one option on the puzzle box (e.g. deciduous foliage associated with the right-hand side, buffalo worm reward on the top left panel). It was only later that I realised this was not the case. It is also not clear what the blue font means. It would be helpful if the figure could clearly indicate which is the optimal (highest value) choice for immigrants.

 In panel B, I wasn't sure why there were two symbols (+ or -) in each quadrant. E.g. if each quadrant represents a prediction about whether immigrants would adopt the resident solution, what does +- mean? Are the colours significant here (e.g. do they relate to the red and blue of the deciduous/pine foliage, or to door colour?). 

L130: It's interesting that the immigrants seemed to settle in immediately with no stress and were not subject to aggression by residents. To put the experiment into context, it would be useful if the introduction could tell us a bit more about immigration events in the wild. Do immigrant birds immediately start joining in social foraging events, and are they left in peace, as in the experiment? This has important implications for the potential to observe and learn from residents.

L155: "although this did not meet our criterion for significance" - what is the criterion for significance?

L182: "When immigrants were socially influenced in the Es; Pa condition, they were estimated to be more influenced by observations of the lower payoff solution by fellow immigrants." If immigrants were more influenced by other immigrants, then why do they still appear to switch to the resident preference in Fig 2? Greater clarity on the role of individual learning would help here. 

L192: Similarly, the statement that "when environments were symmetric, immigrants… were more likely to ignore higher payoff solutions produced by residents" initially seems at odds with Fig 2. In the figure, it seems that in the Es, Pa condition immigrants still ended up choosing the resident solution around 60% of the time (Es; Pa = 0.676 on line 143). I can see that they are "more likely" to ignore the resident solution relative to the Ea, Pa condition (although the difference isn't huge) but overall it seems they are still pretty keen to switch to the resident solution when the environments are symmetrical. A clear explanation of how this comes about (through individually learning the optimal solution?) would be helpful.

Results - general query. Given the authors' previous work on conformist social learning, I was wondering throughout whether and how the relative numbers of resident solvers and immigrants matter. In some groups only 3 non-tutor residents learned to solve the task, whereas in others all 7 non-tutors were solvers. As immigrants will also see the attempts of fellow immigrants (plus a small number of non-tutor solutions by residents), the proportion of resident tutor solutions they are exposed to may be rather variable. Do the models account for this, and if so to what degree are immigrants affected by (a) the absolute number of times they witness the resident tutor solution, (b) the number of individuals using the resident tutor solution or (c) the proportion of solves that use the resident tutor solution vs the other option?

L212: Typo - "can be exhibit"

L226: "a payoff bias is less costly" than what?

L249-252: It would be useful to present some of this information on natural patterns of migration in the Introduction, to give context to the study. For example, until I got to this point, I was wondering whether it is really realistic to think that great tits migrate between totally different habitat types.

Reviewer #2:

The current study reports a social learning experiment in wild-caught great tits, examining how the social and physical environment influences social learning in immigrants. After learning a puzzle box solving behaviour, birds were experimentally relocated into a population with residents which had learned an alternative solution. This new population was housed in either a symmetric or asymmetric physical environment, and the puzzle box solution either produced an identical or higher-payoff reward. Immigrants learnt more rapidly when both the physical environment and payoff changed.

The study is a valuable contribution to the literature, particularly as it teases apart the influence of social and physical environment on social learning in immigrants, something which is highly challenging in wild populations. This type of experimental study in taxa in which experimental relocation is feasible and ethically acceptable is of great value in informing more observational work in primates or large mammals, with which such studies would be impossible to achieve. I therefore think it will be of great interest to a broad range of researchers.

Overall I found the study design very thorough and well-focused on the authors' questions. The manuscript is very well written and cites the appropriate literature. I therefore have only very minor comments.

Minor comments:

Typo in line 58 "social learning individuals should not copying all information"

Line 70: "male immigrants employed a pay-off biased learning strategy that overrode their model bias, while philopatric females continued to prefer to learn from other dominant females" I believe that in this study both juvenile and adult males were tested, so results from both resident and immigrant males were combined when finding the sex effect referred to here. I think the authors' point stands but to my knowledge the cited study did not explore whether immigrant adult males differed from resident juveniles in their learning preference (i.e. the found result suggests a payoff bias in the dispersing sex, but not necessarily an effect of migration) so this should perhaps be rephrased. This comes up again in line 217 - it may well be that the result found in the original study was driven by immigrant males, but to my knowledge it was not analysed in a way that would allow this conclusion to be drawn as adult and juvenile male data were combined.

Line 262-263: "we further show that these changes in learning strategies are not due to underlying life-history state, e.g. tied to hormonal changes during dispersal". This is certainly the case within the current experiment, but I'm not convinced this conclusion can be generalised from the current study to rule out this causal mechanism as a contributing factor in other species? I.e. the fact that these learning strategies can be elicited via experimental immigration events alone does not exclude the possibility that hormonal changes during dispersal contribute in other observations. Perhaps these would have an additive effect (an increased attention to social information, or more rapid learning, for example).

Figure 2 is a very clear communication of the results (I received the updated version prior to review).

The statistical analysis appears appropriate and thorough. 

Reviewer #3:

The value and use of social information is an issue that is highly topical with a broad reach in terms of audience who should be interested in the data presented here. I think the experiment is clearly explained, conducted well, and the data analysed appropriately. I do, however, have a question or two regarding the rationale, the appropriateness of the experimental design for this question, and the interpretation of the data, which leave me thinking that the authors are missing some key parts to this story. 

L40: nothing further is said here about the resident responses to immigrants and there is only a very little further in the text. This appears to be a strong and rapid result, albeit not long-lived but I do not see any explanation. I am not sure how the data shown in Figure S4 map to the data in Fig 2? 

I missed a very clear hypothesis. There is a prediction that the impact of spatial variation should be greater than that of temporal variation on the probability of learning from others but I am not sure that the authors have clearly explained why this would be the case? Furthermore, in the experiment that has been conducted there is clear spatial variation (here a change in some of the physical aspects of the environment), which then sets the scene for attention to change in food reward/the behaviour of the other individuals. I think the other cue that has changed is the identity of the individuals to which the immigrants are exposed (as the authors also finally suggest in lines 252-253). It appears that by being moved to a different environment (the asymmetric environment), this change in physical appearance is big enough that the birds have been cued to look for difference. They then immediately pay attention to the residents, learning rapidly (stability within some 10 trials) in both instances to switch or stick with their response to the puzzle (ie the response learned in the previous environment). The move to the 'same'/symmetric environment does not, however, alert the birds with the asymmetric payoff in the same way so they pay less attention to the residents. Over time, however, they begin to do so, and then begin to switch their response. This doesn't seem to me to be a test of spatial versus temporal information but rather of spatial versus social information (here identity of new group mates). Possibly because these birds do move around with a flexible number and identity of flock mates, this information is simply less salient to them. I must have missed something and may well be using the wrong language (e.g. words like attention) to describe the data. 

Paying attention to flock mates is sometimes also something the residents also do. The data show that when the residents are getting better food, they ignore the behaviour of the immigrants (upper 2 panels in Fig 2), but they pay immediate attention (and copy) the behaviour of the immigrants in the alternative payoff situation. The temporal component is that the spatial change occurs before the availability of information about food reward but isn't really testing the importance of one up against the other as, as I understand it, the authors seem to suggest they have done. Clarifying this would be hugely helpful. 

I think it would also help put the interpretation of the data into a broader context that was about the mechanistic basis of learning and not just how information might be put to use once an animal has acquired it. Here the authors explain their data in the context of high uncertainty but it is not clear how they have tested high temporal uncertainty. 

Details:

L219: is there spatial variability in the payoff cues?

Any idea why the immigrants in the asymmetric payoff conditions never get above 75%?

Overall, I think the writing could do with some editing to make it more accessible to the audiences I think the authors might like to reach. Currently there is a serious expectation that the readership is very familiar with the language and background: from assuming that Rogers' (I think the apostrophe in the text might be in the wrong place?) paradox is known outside the social learning crowd to the discussion in lines 240-244. If the writers give the readers less work to do they will feel more clever, and be more impressed by the data.

---

## [Editor Report · Decision Letter 2]

17 Sep 2024

Dear Michael,

Thank you for your patience while we considered your revised manuscript "Immigrant birds use payoff biased social learning in spatially variable environments" for publication as a Short Report at PLOS Biology. This revised version of your manuscript has been evaluated by the PLOS Biology editors and the Academic Editor.

Based on our Academic Editor's assessment of your revision, we are likely to accept this manuscript for publication, provided you satisfactorily address the following data and other policy-related requests.

IMPORTANT - please attend to the following:

a) We're struggling with the Title a bit - yours is concise and accurate, but we think it may be hard for the broader reader to parse. The meaning of "payoff-biased" is particularly opaque. Maybe change it to something like "Immigrant birds learn from payoffs obtained by local residents in spatially variable environments" or "Immigrant birds learn from socially observed differences in payoffs in spatially variable environments"? Also am I right in thinking that "spatially variable" means when the new environment differs from the old? If so, it may be clearer to express that here ("...when their new environment differs"?). Another option might be to include mention of the experiment ("A cultural diffusion experiment reveals that social learning by immigrant birds is payoff-biased when environments are spatially variable"), but including some of the features mentioned previously. Happy to discuss this by email (rroberts@plos.org) if you wish.

b) In the Abstract, you say "with captive wild great tits (Parus major)"; this seems an oxymoron. Perhaps "with wild great tits (Parus major) in captivity" may be less confusing?

c) Please address my Data Policy requests below; specifically, we need you to supply the numerical values underlying Figs 2, 3, 4, S1, S2, S3, S4, S5, S6, either as a supplementary data file or as a permanent DOI’d deposition. I note that you already have an associated Edmond deposition (https://doi.org/10.17617/3.FXC12W), but that contains a single ~7-Gb zipped folder and is likely to be inaccessible to many users. Perhaps the numerical values could be made available as a supplemetary spreadsheet (S1 Data) or as a small extra file in the Edmond deposition?

d) Please cite the location of the data clearly in all relevant main and supplementary Figure legends, e.g. “The data underlying this Figure can be found in S1 Data and https://doi.org/10.17617/3.FXC12W"

e) Please make any custom code available, either as a supplementary file or as part of your data deposition.

We expect to receive your revised manuscript within two weeks. 

*Published Peer Review History*

*Press*

Sincerely,

Roli

Roland Roberts, PhD

Senior Editor

rroberts@plos.org

PLOS Biology

DATA POLICY:

Regardless of the method selected, please ensure that you provide the individual numerical values that underlie the summary data displayed in the following figure panels as they are essential for readers to assess your analysis and to reproduce it: Figs 2, 3, 4, S1, S2, S3, S4, S5, S6. NOTE: the numerical data provided should include all replicates AND the way in which the plotted mean and errors were derived (it should not present only the mean/average values).

CODE POLICY

DATA NOT SHOWN?

---

## [Editor Report · Decision Letter 3]

24 Sep 2024

Dear Michael,

Thank you for the submission of your revised Short Reports "Immigrant birds learn from socially observed differences in payoffs when their environment changes" for publication in PLOS Biology. On behalf of my colleagues and the Academic Editor, Lars Chittka, I'm pleased to say that we can in principle accept your manuscript for publication, provided you address any remaining formatting and reporting issues. These will be detailed in an email you should receive within 2-3 business days from our colleagues in the journal operations team; no action is required from you until then. Please note that we will not be able to formally accept your manuscript and schedule it for publication until you have completed any requested changes.

Sincerely, 

Roli

Senior Editor

PLOS Biology

rroberts@plos.org